# PFO: Optimizing binary Preference Alignment from a Probability Flow Perspective

## Abstract

In contrast to pairwise optimization methods, binary preference alignment algorithms do not process sample pairs jointly, which may lead to suboptimal probability distributions. Under the influence of the squeezing effect, probability mass flowing out of negative samples may diffuse into neutral regions, while the mass absorbed by positive samples might originate from such neutral areas, resulting in insufficient penalty for negative responses. To address this issue, we propose the PFO (Probability Flow Optimization) algorithm. The algorithm dynamically evaluates the probability of sample generation and systematically optimizes the transfer of probability mass by reweighting samples to encourage flow from negative to positive distributions. Comprehensive experiments and analysis on the general-purpose benchmarks MT-Bench and AlpacaEval 2 demonstrate the algorithm's effectiveness. Furthermore, experiments on recommendation domain datasets show that the method effectively applies to sparse feedback scenarios, confirming the algorithm's broad applicability. Our work offers new insights into improving binary preference alignment from the perspective of probabilistic flow.

## 1 Introduction

Using preference data to fine-tune large language model(LLM) is crucial to align them with human preferences and make them more helpful and less harmful (Bai et al., 2022). In recent years, preference alignment training for LLM has been extensively explored. The researchers trained a reward model on human preference data to evaluate the responses of LLM and first proposed the Reinforcement Learning from Human Feedback (RLHF) paradigm to align large language models with human preferences (Ouyang et al., 2022). Subsequently, the research community is steadily expanding its focus on employing reinforcement learning algorithms to train an LLM policy that maximizes the quality score of its responses to specific prompts, thus aligning the generated content with human preferences (Yue et al., 2025; Yu et al., 2025a; Yan et al., 2025).

However, due to the difficulty of modeling quality scoring rewards, it is challenging to achieve both accuracy and generalization across multiple tasks. To address these issues, off-line preference fine-tuning methods have been proposed, such as DPO (Rafailov et al., 2023), IPO (Azar et al., 2024), and SIMPO (Meng et al., 2024). These methods utilize human-annotated offline preference data and replace reinforcement learning with supervised learning. They typically utilize paired labeled data and are trained by maximizing the difference between the likelihood of positive and negative samples. Although Pairwise methods have achieved excellent results, they come with extremely high costs in data acquisition because online feedback is often directed at a displayed reply and either likes or dislikes it. Consequently, it is crucial to find ways to leverage cheaper single-label training data to better align models with human preferences. In this vein, KTO (Ethayarajh et al., 2024) proposed a binary preference loss that learns from single-label data without requiring pairwise comparisons.

Although the KTO algorithm enables binary preference alignment training, its inherent binary preference labeling means each training iteration can only operate in one direction: either encouraging positive sample likelihood or penalizing negative sample likelihood. This mechanism may be more susceptible to the squeezing effect (Ren & Sutherland, 2024). Specifically as Figure 1 shown, if the probability of negative sample is very low during optimization, the probability mass released by them may not be effectively transferred to positive sample, but instead flow to other non-target

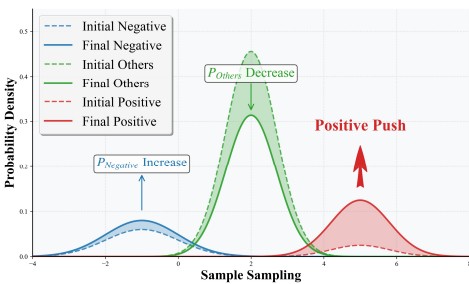 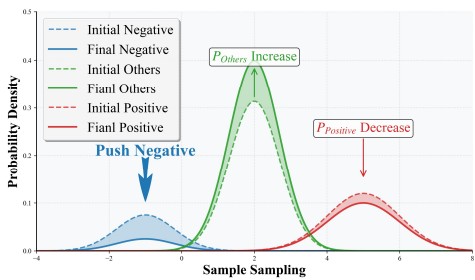

(a) The squeezing effect of positive training.   (b) The squeezing effect of negative training.

Figure 1: The squeezing effect when training with extremely low probability samples. When training on samples with low probability, the squeezing effect may lead to insufficient encouragement for positive samples and insufficient punishment for negative samples.

samples, resulting in an insufficient increase in the probability of positive sample. Conversely, if the probability of positive sample is low during optimization, the probability mass absorbed may mainly come from non-target samples, rather than effectively reducing the probability of negative sample, thereby weakening the suppression effect on negative sample.

In this paper, we propose a binary preference alignment algorithm called PFO (Probability Flow Optimization) based on probability flow analysis to address the probability flow issue in the aforementioned squeezing effect. The essence of PFO is to dynamically adjust the weights of different samples in the batch during gradient backpropagation based on the model's current likelihood, thereby mitigating the degree of model squeezing. This encourages the model to direct probability flows from negative samples to positive samples, preventing the phenomenon where positive sample probabilities are not effectively increased and negative sample probabilities are not effectively decreased. Furthermore, we provide additional theoretical proof that our optimization objective is a lower bound for the Rényi $\mathcal{D}_2$ divergence.

Moreover, based on the trade-off between the positive and negative distributions of the squeezing effect, we propose PFO+, a variant of PFO that performs better in scenarios where positive data require more protection (e.g., recommendation systems). In this case, PFO+ degenerates into the lower bound of the KL divergence in KLDO (Haldar et al., 2025), but its performance remains superior to the original KLDO due to the influence of weight constraints.

Finally, we experimentally demonstrate that the PFO algorithm outperforms KTO and KLDO across multiple models on the MT-Bench (Zheng et al., 2023) and AlpacaEval 2 (Li et al., 2023) benchmarks. Furthermore, we validate that the PFO variant PFO+ demonstrates superior preference understanding and recommendation capabilities compared to KTO and KLDO on the open-source recommendation datasets MoviesLens-25M (Harper & Konstan, 2015) and Goodreads (Wan et al., 2019).

## 2 RELATED WORK

Reinforcement Learning from Human Feedback (RLHF) is a key technology that enhances the response quality of large language models by aligning them with human preferences. Its training process consists of three main stages. The model is first initialized by supervised fine-tuning (Zhou et al., 2023; Javaheripi et al., 2023). Then a reward model is trained based on manually labeled preference data, to output a scalar reward signal for the generated responses of the policy (Wang et al., 2024; Liu et al., 2024; Yu et al., 2025b). Finally, reinforcement learning algorithms such as Proximal Policy Optimization (PPO) (Schulman et al., 2017) are employed to optimize the policy. In recent years, more stable alternatives have emerged, including GRPO (Shao et al., 2024) and VinePPO (Kazemnejad et al., 2024). Although RLHF demonstrates significant advantages in domains such as mathematical reasoning (Team et al., 2025) and safety alignment (Bai et al., 2022), it remains

challenging to deploy across many domains compared to traditional supervised learning due to its more complex training process and the requirement for precise rewards.

Due to the complex implementation and poor training stability of RLHF, researchers have begun to promote off-line preference alignment methods based on labeled data. Rafailov et al. (2023) proposed to implicitly define the reward function based on the likelihood of the policy, bypassing the construction of an explicit reward model. Inspired by them, a large number of offline direct preference alignment methods have been proposed, such as SIMPO (Meng et al., 2024), SIMPER (Xiao et al., 2025), and EXO (Ji et al., 2024a). Although these methods have proven effective, they typically rely on paired preference-labeled data. To address this limitation, Ethayarajh et al. (2024) proposed the KTO algorithm, which decomposes paired preference training into binary preference training requiring only samples labeled as "good" or "bad". Mao et al. (2024) proposed an algorithm based on fine-tuning principles to establish stable bidirectional negative feedback during optimization. Haldar et al. (2025) introduced a preference optimization method for safety alignment from a divergence estimation perspective. However, the aforementioned approach overlooked the issue of probability flows deviating from the intended direction due to training data asymmetry, potentially leading to limitations such as insufficient positive reinforcement or inadequate negative punishment.

## 3 THE PROPOSED METHOD

### 3.1 BACKGROUND

**Notations.** We formalize the preference dataset as $\mathcal{D} = \{(x, y_w, y_l)\}$, where each data point consists of an input prompt $x$ and its corresponding response pair $\mathbf{y} = (y_w, y_l)$ generated by a language model, with $y_w$ denoting the chosen response and $y_l$ the rejected response. For binary preference alignment, each prompt $x$ contributes exactly only one response, thus inducing partitioned datasets $\mathcal{D}^+ = \{(x, y_w)\}$ and $\mathcal{D}^- = \{(x, y_l)\}$. The optimization objective involves learning parameters $\theta$ of the policy $\pi_\theta(\mathbf{y}|x)$, which is initialized from a reference policy $\pi_{ref}$, to achieve human preference alignment in generated responses.

**DPO.** Rafailov et al. (2023) proposed a preference alignment framework that does not require explicit reward modeling. It utilizes the Bradley-Terry (Bradley & Terry, 1952) preference model to derive a closed-form solution for the optimal policy and establishes a mapping relationship from policy parameters to implicit rewards, i.e., $r_\theta(x, y) = \beta \cdot \log \left[\pi_\theta(y \mid x)/\pi_{\text{ref}}(y \mid x)\right]$. Ultimately, it directly optimizes the policy by maximizing the likelihood of preference data.

$$\mathcal{L}_{\text{DPO}}(\theta) = -\mathbb{E}_{(x,y_w,y_l)\sim\mathcal{D}} \left[\log \sigma \left(r_\theta\left(x, y_w\right) - r_\theta\left(x, y_l\right)\right)\right]. \tag{1}$$

**KTO.** KTO (Ethayarajh et al., 2024) is a preference alignment method for binary signals. Inspired by prospect theory in economics, KTO decouples the losses of positive and negative preferences to enable training with single-label feedback data. The KTO loss with a reference non-negative constant $z_0$ is defined as follows.

$$\mathcal{L}_{\text{KTO}}(\theta) = \mathbb{E}_{x,y_w\sim\mathcal{D}^+} \left[1 - \sigma \left(r_\theta(x, y) - z_0\right)\right] + \mathbb{E}_{x,y_l\sim\mathcal{D}^-} \left[1 - \sigma \left(z_0 - r_\theta(x,y)\right)\right] \tag{2}$$

**KLDO.** Haldar et al. (2025) demonstrated that mainstream alignment algorithms such as DPO and KTO essentially correspond to variational estimates of TV divergence. They constructed the objective function KLDO using the DV representation of KL divergence, thereby achieving a clearer separation between safe response distributions and harmful response distributions.

$$\mathcal{L}_{\text{KLDO}}(\theta) = - \mathbb{E}_{x,y_w\sim\mathcal{D}^+} r_\theta(x,y) + \ln \mathbb{E}_{x,y_l\sim\mathcal{D}^-} e^{r_\theta(x,y)} \tag{3}$$

**Squeezing Effect.** Ren & Sutherland (2024) conducted a dynamic analysis of each step during the training process and proposed an adverse phenomenon termed the "squeezing effect". They observed when large negative gradients are applied to already-unlikely outcomes, peak probabilities may be amplified while simultaneously other probabilities, including those of positive samples may be suppressed.

### 3.2 SQUEEZING EFFECT EXTENDED CLAIMS

Inspired by the squeezing effect (Ren & Sutherland, 2024), we continue with its proof approach and extend three claims in binary preference alignment:

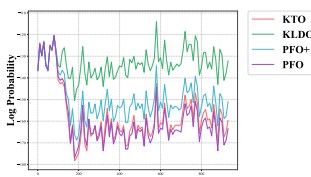 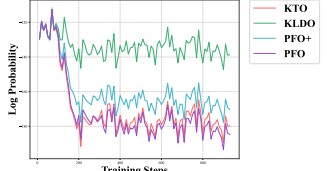 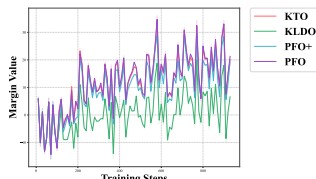

(a) The log probability of chosen during training.

(b) The log probability of rejected during training.

(c) The log probability margin during training.

Figure 2: The training dynamics during training of KTO, KLDO, PFO and PFO+ on the Gemma2-9B-Instruct (Team et al., 2024). We can observe that in the binary preference alignment training, the log probability of positive samples also shows a decreasing trend. And the PFO algorithm we proposed achieves the most significant improvement in the margin of rejected and chosen.

---

**Takeaway  Squeezing effect trend in binary preference alignment**

*For Negative Training:*
**Claim 1.** The probability of negative sample $p_y$ will decrease, while the probability of peak $p_{i*}$ will increase.
**Claim 2.** After negative training, $p_i$ with smaller $p$ tends to decrease and vice versa.
**Claim 3.** When the probability of negative sample is extremely small and a peak occurs, except the peak probability $p_{i*}$ increasing, all other probabilities $p_i$ will decrease.

*For Positive Training:*
**Extended Claim 1.** The probability of positive sample $p_y$ will increase, while the probability of peak $p_{i*}$ will decrease.
**Extended Claim 2.** After positive training, $p_i$ with smaller $p$ tends to increase and vice versa.
**Extended Claim 3.** When the probability of positive sample is extremely small and a peak occurs, except the peak probability $p_{i*}$ decreasing, all other probabilities $p_i$ will increase.

---

The proof is in Appendix A.1. It is obvious that both **Claim 3** and **Extended Claim 3** are unacceptable. When the **Claim 3** trend occurs, all probabilities are pushed toward the peak, which is detrimental to encouraging positive samples. Similarly, when the **Extended Claim 3** trend occurs, probabilities flowing out from the peak tend to scatter in all directions, which is also detrimental to penalizing negative samples.

### 3.3 THE LEARNING OBJECTIVE OF PFO

In this section, we describe in detail the optimization objective PFO that we proposed. The core idea is that we believe greater weight should be applied to samples with higher probabilities in order to reduce the squeezing effect, thereby driving the probability to maximize outflow from negative samples and flow into the positive.

As shown in Figure 2a, our binary preference alignment training experiments exhibit similar phenomena to those reported by Rafailov et al. (2024) and Razin et al. (2024) with a balanced number of positive and negative samples and coefficient. The logits of rejected samples decrease during training, and we also observe a corresponding decrease in the logits of chosen samples. This suggests that in addition to the direct optimization effect of positive and negative samples, the negative squeezing effect dominates the training dynamics.

To more effectively guide model optimization, we propose to enhance the weight coefficients of samples with larger $r_\theta$. Specifically, this objective is achieved through the following optimization function:

$$\mathcal{L}_{\text{PFO}}(\theta) = -log \mathop{\mathbb{E}}_{x,y_w \sim \mathcal{D}^+} e^{\sigma(r_\theta(x,y))} + log \mathop{\mathbb{E}}_{x,y_l \sim \mathcal{D}^-} e^{\sigma(r_\theta(x,y))} \tag{4}$$

Initially, we considered directly using $\pi_\theta$ as the reward function $r_\theta$ to provide an intuitive weight. However, as shown in Figure 3, the weight variation range under this setting was limited, not achieving the expected reweight effect. We need to add some margin to the design of $r_\theta$.

Based on the observation above mentioned that the negative squeezing effect is dominant, we propose the following inference: For negative samples, if $\pi_\theta - \pi_{ref}$ is large, it indicates that the current policy is not penalizing enough, and therefore the penalty should be further strengthened. As for positive samples, the large difference indicates that the probability was less affected by the squeezing effect, and according to Claim 2, should have a higher likelihood value $\pi_\theta(y|x)$. Therefore, we ultimately chose to use the difference between $\pi_\theta$ and $\pi_{ref}$ as the reward function $r_\theta$.

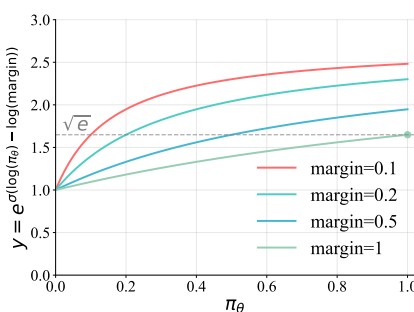

Figure 3: If $\pi_\theta$ is designed directly as $r_\theta$, i.e., $log(margin) = 0$, the weight coefficient will have a very small range $[1, \sqrt{e}]$

### 3.4 ANALYSIS AND DISCUSSION

In this section, we provide an analysis of the PFO optimization objective, including gradient analysis and divergence analysis, to further understand how our PFO mitigates the squeezing effect.

#### 3.4.1 GRADIENT ANALYSIS

We compare the gradient analysis of the optimization objectives KTO, KLDO, and PFO. First, we analyze the gradient of the proposed PFO and investigate how it alleviates the squeezing effect:

$$\nabla_\theta \mathcal{L}_{\text{PFO}}(\theta) = -\frac{\underset{x,y_w \sim \mathcal{D}^+}{\mathbb{E}} \nabla_\theta \sigma(r_\theta) e^{\sigma(r_\theta)}}{\underset{x,y_w \sim \mathcal{D}^+}{\mathbb{E}} e^{\sigma(r_\theta)}} + \frac{\underset{x,y_l \sim \mathcal{D}^-}{\mathbb{E}} \nabla_\theta \sigma(r_\theta) e^{\sigma(r_\theta)}}{\underset{x,y_l \sim \mathcal{D}^-}{\mathbb{E}} e^{\sigma(r_\theta)}} \tag{5}$$

In the proposed PFO method, the optimization weights of positive and negative samples are dynamically adjusted based on their values of $\sigma(r_\theta)$, i.e., $\sigma(\beta \cdot \log[\pi_\theta(y \mid x)/\pi_{\text{ref}}(y \mid x)])$. This ensures that when the difference between $\pi_\theta$ and $\pi_{ref}$ is large, the contribution of the gradient in a batch increases, thus alleviating the squeezing effect and protecting the probability of flowing effectively from negative to positive. Specifically, the $\sigma(\cdot)$ function applies a non-linear mapping to the original preference score $r_\theta$, with the core objective of preventing the numerical dominance effect caused by certain samples with excessively large $r_\theta$ values when $r_\theta$ is used directly as the exponential weight, thereby overwhelming the contributions of samples with lower $r_\theta$ values. It is worth noting that the introduced sigmoid function not only effectively alleviates the sample drowning problem, but also helps stabilize the gradient dynamics during training.

For comparative analysis, we examine the gradient characteristics of KTO and KLDO:

$$\nabla_\theta \mathcal{L}_{\text{KTO}}(\theta) = -\underset{x,y_w \sim \mathcal{D}^+}{\mathbb{E}} \nabla_\theta \sigma(r_\theta - z_0) + \underset{x,y_l \sim \mathcal{D}^-}{\mathbb{E}} \nabla_\theta \sigma(r_\theta - z_0) \tag{6}$$

$$\nabla_\theta \mathcal{L}_{\text{KLDO}}(\theta) = -\underset{x,y_w \sim \mathcal{D}^+}{\mathbb{E}} \nabla_\theta r_\theta + \frac{\underset{x,y_l \sim \mathcal{D}^-}{\mathbb{E}} \nabla_\theta\, r_\theta e^{r_\theta}}{\underset{x,y_l \sim \mathcal{D}^-}{\mathbb{E}} e^{r_\theta}} \tag{7}$$

Gradient analysis indicates that KTO's gradient does not incorporate an adaptive weight adjustment mechanism for each sample, but instead performs an indiscriminate expectation estimation of the gradients for all samples. Under this approach, training low-probability samples may trigger the situations described in **Claim 3** and **Extended Claim 3**, resulting in insufficient protection of positive sample probabilities and insufficient punishment of negative sample probabilities.

Gradient analysis of KLDO reveals that it also introduces an adaptive weighting mechanism based on $r_\theta$ in the negative penalty term to dynamically adjust the contribution of samples within a batch, thus mitigating the squeezing effect. However, KLDO has two key flaws. First, it lacks a corresponding contribution adjustment mechanism for positive samples, which may lead to insufficient

preventive optimization for the trend described in **Extended Claim 3**, which would weaken the penalty for negative samples. Secondly, it implements weight distribution through an exponential function $e^{r_\theta}$ of the unbounded variable $r_\theta$. When samples with extremely large $r_\theta$ values exist in a batch, the exponential amplification effect can lead to significant weight differences between samples, significantly suppressing or even overwhelming the contributions of some samples.

### 3.4.2 DIVERGENCE ANALYSIS

Next we will illustrate the relationship between the proposed PFO and the divergence between positive and negative samples from a distribution perspective.

**Theorem 3.1** *The maximization objective of PFO constitutes a variational lower bound for the the Rényi $\alpha$-divergence of order $\alpha = 2$ between the positive and negative samples.*

$$\mathcal{D}_2(\mathcal{P} \parallel \mathcal{Q}) \triangleq \log \sum_x \frac{P^2(x)}{Q(x)} = \sup_T log \mathop{\mathbb{E}}_{v \sim P} e^{T(v)} - log \mathop{\mathbb{E}}_{v \sim Q} e^{T(v)} \tag{8}$$

The proof is provided in Appendix A.2. It's worth noting that when measuring the difference in probability mass distributions of language models, the degree of fit in high-probability regions should be prioritized. Compared to the KL divergence, the Rényi $\mathcal{D}_2$ divergence characterizes the difference in the second-order moments between distributions. The quadratic term $P^2(x)$ in its definition has a significant suppressive effect on the contribution of low-probability events. This property allows the $\mathcal{D}_2$ divergence to more precisely reflect differences in high-probability regions (i.e., the main distribution), while being more robust to small changes in low-probability regions.

To alleviate the sample drowning problem, we introduced a sigmoid function as a nonlinear mapping in the parameterization of $T(v)$. However, the constraints imposed by this function on the output values may make the theoretical upper bound of PFO unattainable in practice, which may affect the tightness of the upper bound. We demonstrated that the resulting variational gap gradually decreases during training until the optimization objective approaches the upper limit of the parameterization capability. The upper bound of this gap is on the order of $O(\log \sqrt{\mathcal{D}_2(\mathcal{P} \| \mathcal{Q})})$. The variational gap decreases as the discrepancy between the positive and negative datasets diminishes. The proof is in Appendix A.3. We conducted multiple hyperparameter experiments (see Section 4 for details) to examine potential training instability arising from this issue, and found that performance remained stable across multiple trials. More theoretical optimization and analysis will be the focus of future research.

### 3.4.3 PFO POSITIVE

As mentioned above, PFO aims to mitigate positive and negative squeezing effects in preference alignment by applying adaptive weights to positive and negative training samples, respectively. Specifically, negative weight adjustment mitigates the issue of weakened positive sample rewards caused by excessively low negative sample probabilities, while positive weight adjustment alleviates the problem of insufficient negative sample penalties resulting from excessively low positive sample probabilities. However, in certain sparse feedback binary preference alignment scenarios, such as recommendation systems, the model prioritizes fully rewarding and protecting positive samples. Due to individual variations, negative samples may not require strict penalties. Therefore, we introduce an adaptive weight adjustment mechanism specifically for negative samples and propose a variant of PFO, PFO+ (PFO Positive).

$$\mathcal{L}_{\text{PFO+}}(\theta) = - \mathop{\mathbb{E}}_{x,y_w \sim \mathcal{D}^+} \sigma(r_\theta(x,y)) + log \mathop{\mathbb{E}}_{x,y_l \sim \mathcal{D}^-} e^{\sigma(r_\theta(x,y))} \tag{9}$$

This optimization objective formally resembles the KLDO algorithm, with the core distinction being the introduction of the sigmoid function, which effectively prevents sample drowning and gradient explosion during training.

## 4 EXPERIMENTS

We designed and conducted two types of experiments to evaluate the effectiveness of the proposed method. The first type of experiment focuses on the evaluation of multi-round dialogue capabilities.

We use a completely open-source and available pipeline to train the model and conduct a comprehensive evaluation and comparative analysis on benchmarks such as MT-Bench (Zheng et al., 2023). The second type of experiment targets a specific scenario of binary alignment, taking the recommendation system experiment as an example. We trained the recommendation system model on the open-source datasets, aiming to verify the performance of the proposed PFO algorithm and its variant PFO+ in a real-world application environment.

## 4.1 DIALOGUE EXPERIMENT

### 4.1.1 EXPERIMENTS SETUP

Table 1: AlpacaEval 2 (Li et al., 2023) and MT-Bench (Zheng et al., 2023) results under the four settings. LC and WR denote length-controlled and raw win rate, respectively. Our PFO and PFO+ can achieve better performance across various settings. PFO+ is a variant of PFO that only focuses on increasing the positive probability.

| Benchmark | Method | Llama3-8B | | Gemma2-9B | | Avg |
| | | Instruct | Base | Instruct | Base | |
| --- | --- | --- | --- | --- | --- | --- |
| MT-Bench | baseline | 7.26 | 5.94 | 7.76 | 6.27 | 6.81 |
| | KTO | 7.53 | 6.57 | 8.00 | **7.19** | 7.32 |
| | KLDO | 7.28 | 6.36 | 7.93 | 7.11 | 7.17 |
| | PFO+ | **7.72** | 6.62 | 7.89 | 7.03 | 7.32 |
| | PFO | 7.62 | **6.88** | **8.13** | 7.00 | **7.41** |
| AlpacaEval 2 (WR/LC) | baseline | 32.72/34.11 | 2.36/3.78 | 36.02/47.06 | 4.71/7.14 | 18.95/23.02 |
| | KTO | 36.69/39.73 | 11.17/15.68 | 44.92/50.25 | 19.83/29.00 | 28.15/33.67 |
| | KLDO | **38.11/41.02** | 6.14/9.32 | 43.84/50.43 | 8.12/12.70 | 24.05/28.37 |
| | PFO+ | 32.72/34.11 | 12.25/16.49 | 44.05/**52.41** | 18.31/27.38 | 26.83/32.60 |
| | PFO | 36.96/40.41 | **14.35/20.56** | **45.06**/50.57 | **20.10/31.01** | **29.12/35.64** |

**Models and Datasets.** We performed alignment with two families of opensource models, Llama3-8B-Base/Instruct (Dubey et al., 2024) and Gemma2-9B-Base/Instruct (Team et al., 2024).

For the **Base** setup, we followed the training pipeline of SIMPO (Meng et al., 2024) and Zephyr (Tunstall et al., 2023). We first trained the base model on theUltraChat-200k dataset (Ding et al., 2023) to obtain an SFT model. We then performed binary preference alignment on the UltraFeedback dataset (Cui et al., 2023)using the SFT model as the starting point.

For the **Instruct** setup, we use off-the-shelf instruction-tuned models, i.e., meta-llama/Meta-Llama-3-8B-Instruct (Dubey et al., 2024) and google/gemma-2-9b-it (Team et al., 2024). In order to obtain better training data, we reused the training datasets princeton-nlp/llama3-ultrafeedback-armorm and princeton-nlp/gemma2-ultrafeedback-armorm processed by Meng et al. (2024). For each prompt in UltraFeedback dataset (Cui et al., 2023), they generated 5 responses using the SFT model. These 5 responses are then scored using llm-blender/PairRM (Jiang et al., 2023), and the response with the highest score is selected as chosen and the response with the lowest score is selected as rejected.

**Evaluation benchmarks.** We primarily evaluate our models using two of the most popular open-source instruction-following benchmarks: MT-Bench (Zheng et al., 2023) and AlpacaEval 2 (Li et al., 2023). These benchmarks assess models' multi-dimensions conversational capabilities across multiple topics and have been widely adopted by the community. MT-Bench covers 80 questions across 8 categories. AlpacaEval 2 contains 805 questions from 5 datasets. For MT-Bench, we report the average MT-Bench score of models evaluated by GPT-4o. For AlpacaEval 2, we report the raw win rate (WR) and length-controlled win rate (LC) (Dubois et al., 2024) of models evaluated against GPT-4-Preview-1106 using GPT-4o as the judgement. The LC metric is specifically designed to be robust to the effects of model length.

**Baselines.** We compare PFO and PFO+ with the binary preference optimization method KTO (Etha-yarajh et al., 2024) and KLDO (Haldar et al., 2025). The coefficient weights for positive and negative training are the same for all methods. Considering that hyperparameter tuning is crucial to

achieve the best performance of the preferred optimization method, we thoroughly tuned the hyper-parameters for each baseline and reported the best performance. More details of baselines and the hyperparameter search space can be found in Appendix B.1.

### 4.1.2 MAIN RESULT ON BENCHMARKS

**Results On Dialogue Benchmarks**

As shown in the Tabel 1, we report the performance of the evaluated algorithms on the multi-turn dialogue benchmarks MT-Bench and AlpacaEval 2. To facilitate comparative analysis between different algorithms and base models, we calculated the average performance of each algorithm on four different models on both bench-mark test datasets. Our proposed algorithm PFO out-performs KTO and KLDO on both MT-Bench and Al-pacaEval 2. Notably, we also evaluated the variant of PFO, PFO+. As described in Section 3.4.3, PFO+ is less effective than PFO because it only mitigates the squeezing effect of negative training. Nevertheless, by introducing the sigmoid function, PFO+ alleviates the

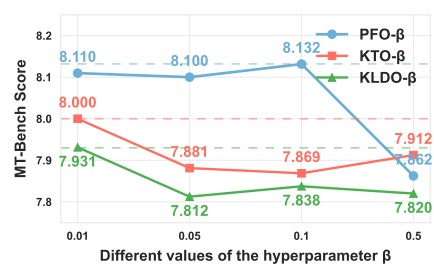

Figure 4: Scores of training models with different hyperparameter $\beta$ on MT-Bench.

sample drowning problem during weight assignment and stabilizes the gradient, resulting in superior results to KLDO on both MT-Bench and AlpacaEval 2.

To verify the conclusions mentioned in Section 3.4.2, a loose upper bound on the PFO objective does not significantly impact training. We report the performance of the Gemma2-Instruct-PFO with different $\beta$ in MT-Bench in Figure 4, which demonstrates that despite the gap between the upper bound and the $\mathcal{D}2$ divergence, the optimization process is robust and stable.

**Likelihood Analysis of Training**

To verify whether the probabilities transfer from negative to positive samples as hypothesized, we report the $\triangle$margin (improvement of the difference in log probabilities of positive and negative samples) for each of the four training methods relative to the original base model. Higher values indicate that the model more effectively rewards positive probabilities and penalizes negative probabilities during training.

Specifically, we randomly sampled 1,000 training samples to test the improvement of the model log-probablity margin before and after training.

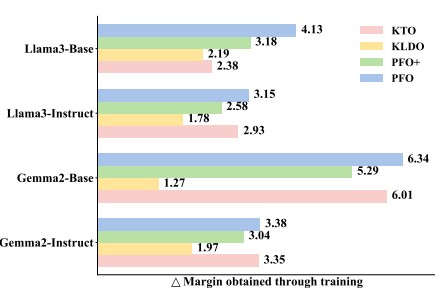

Figure 5: The growth of log probability margin in training samples.

As shown in Figure 5, our proposed algorithm PFO achieves greater improvements in margins across all four models compared to KLDO and KTO.

This demonstrates the effectiveness of the PFO suppression mitigation strategy, which alleviates the increase in negative sample probability and decrease in positive sample probability caused by the squeezing effect during binary preference training. Furthermore, the figure shows that our proposed variant PFO+ achieves a larger margin than KLDO, consistent with the observations in Table 1, indicating that PFO+ outperforms KLDO.

It can also be observed that the log-likelihood margin for PFO+ and KLDO remains lower than that of KTO. We speculate this occurs because, while mitigating the squeezing effect during negative sample training prevents probability decline in positive samples, it simultaneously weakens the direct penalty effectiveness during negative sample training. This results in insufficient punishment for certain low-probability negative sample distributions. In contrast, PFO further penalizes negative sample probabilities by adjusting weights during positive sample training, thereby achieving a larger log-likelihood margin.

## 4.2 RECOMMENDATION EXPERIMENT

**Model and Datasets.** We evaluated a binary preference recommendation system for next-query prediction using user historical queries with the open-source Qwen2.5-Instruct-7B model (Team, 2024). The experiment utilized the MoviesLens-25M (Harper & Konstan, 2015) and Goodreads (Wan et al., 2019). The datasets employ real-world user logs, where training and test partitions are temporally segmented. Query examples are shown in Appendix B.2. Following the data processing approach in Ji et al. (2024b) and Gao et al. (2025), for each user interaction sequence, the terminal element constitutes test data while preceding elements form train data. Rejected samples are queries from other users and different from the chosen sample. Other implementation details are the same as the dialogue experiment configuration in Experiment 4.1.

**Metrics.** We evaluated model performance using HitRate and DivRatio. The HitRate metric is commonly used in recommendation systems, indicating the percentage of items recommended by the model that match those in the ground truth data. The Divratio metric is defined as the proportion of unique recommendations, representing the diversity of the recommendations. We did not filter the results of the model for validity but simply ignored the content in brackets, since we contend that the ability to precisely generate correct item names serves as a key indicator of a model's proficiency.

**Result.** As shown in the Table 2, we report the performance of the evaluated algorithms on the recommendations task on MoviesLens-25M and Goodreads datasets. From the perspective of HitRate, both of our proposed algorithms PFO and PFO+ outperform KTO and KLDO. Furthermore, the improved performance of PFO+ over PFO confirms that in scenarios such as recommendation systems where positive data probability is crucial, PFO+ more effectively preserves the likelihood of positive samples, thereby leading to better results.

Table 2: MoviesLens-25M (Harper & Konstan, 2015) and Goodreads (Wan et al., 2019) results.

| Metric | Model | Goodreads | MovieLens | Avg |
|---|---|---|---|---|
| HitRate (@5/@10) | KTO | 0.020/0.029 | 0.028/0.033 | 0.024/0.031 |
| | KLDO | 0.022/0.023 | 0.029/0.036 | 0.022/0.030 |
| | PFO+ | 0.026/0.038 | **0.033/0.043** | **0.030/0.040** |
| | PFO | **0.033/0.042** | 0.021/0.028 | 0.027/0.035 |
| DivRatio | KTO | 0.1207 | 0.2223 | 0.1715 |
| | KLDO | 0.1423 | 0.1567 | 0.1495 |
| | PFO+ | 0.2149 | 0.1947 | 0.2048 |
| | PFO | **0.2227** | **0.2427** | **0.2327** |

In terms of diversity, both the two proposed methods also exceed KTO and KLDO, indicating that our approach offers greater protection of the overall distribution diversity. This benefit stems from our strategy of redirecting probability mass from negative to positive data, thereby mitigating the squeezing effect and reducing probability loss in non-target distributions. Compared to PFO+, PFO achieves higher diversity, as it incorporates an additional weight adjustment mechanism during positive data training to further alleviate the squeezing effect.

## 5 CONCLUSION

In this paper, we explore the optimization of binary preference alignment methods from a probability flow perspective. We first complement the negative squeezing effect with a positive squeezing effect in our theoretical analysis. Based on this analysis, we propose the PFO and PFO+ algorithms which mitigate the squeezing effect during training and enhance the probability maximization from negative to positive flows. We also provide a theoretical interpretation of PFO, demonstrating that it constitutes a lower bound on the second-order Rényi divergence, which better captures differences in high-probability regions compared to KL divergence. Extensive experiments on widely used benchmarks and datasets show that PFO and its variant significantly outperform the state-of-the-art methods. We hope that our findings offer new insights into the squeezing effect and suggest potential directions for refining binary preference alignment algorithms.

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

## A APPENDIX

### A.1 PROOF OF CLAIMS

*Proof.* For ease of understanding, we continue with the note by Ren & Sutherland (2024) . The ratio of change in the confidence of each token $i$ is represented by Ren & Sutherland (2024) as:

$$\alpha_i \triangleq \frac{p_i^{t+1}}{p_i^t} = \frac{\sum_{j=1}^V e^{z_j^t}}{\sum_{j=1}^V \beta_j e^{z_j^t}}. \tag{10}$$

Where $p_i^t$ denotes the probability of a particular token $i$ while $p_i^{t+1}$ represents the probability after one training step. $z$ denotes the logits of the policy. $V$ denotes the length of the vocabulary. Note that the value expression of $\beta$ depends on whether token $i$ is the trained sample $y$, for $Case1(i = y)$ and $Case2(i \neq y)$:

$$\text{Case 1: } \beta_j = \begin{cases} e^{-\eta'\left(1+p_j^t-p_i^t\right)} & \text{if } j \neq y \\ 1 & \text{if } j = y \end{cases} \text{ ; } \quad \text{Case 2: } \beta_j = \begin{cases} e^{-\eta'\left(p_j^t-p_i^t\right)} & \text{if } j \neq y \\ e^{-\eta'\left(p_j^t-p_i^t-1\right)} & \text{if } j = y \end{cases} \tag{11}$$

Where $\eta'$ is the learning rate, which can be negative if negative gradients are considered during training, and vice versa.

**Extended Claim 1**: For positive sample optimization, denote $p_{i*} = argmax_{i\in[V]\setminus\{y\}}p_i$ as the peak probability other than target sample $y$. In Case 1($\beta \leq 1$), the scaling factor satisfies $\alpha > 1$, inducing increase in $p_i$. In Case 2, for non-target class ($j \neq y$) the condition $p_j \leq p_{i*}$ implies $\beta \geq 1$, while for the target class ($j = y$) $\beta > 1$ holds strictly, collectively yielding $\alpha < 1$ and consequently decrease in $p_i$.

**Extended Claim 2**: Now we observe $p_i'$ and perform a qualitative analysis. Consider the partition of classes into two subsets: $\mathcal{B} = \{j : p_j > p_i'\}$ and $\mathcal{S} = \{j : p_j <= p_i'\}$. In Case 2, as $p_i$ increases, the contribution of terms with $\beta_j > 1$ in $\mathcal{S}$ dominates, increasing the probability that $\alpha < 1$, thus inducing a decrease in $p_i$. Conversely, at lower $p_i$ values, terms with $\beta_j < 1$ in $\mathcal{B}$ become dominant, driving an increase $p_i$.

**Extended Claim 3**: When a peaky target probability $p_{i*}$ emerges, the corresponding logit $z_{i*}$ will also be very large . Under this condition, the value of alpha is dominated by $\beta_{j*}$, i.e., for the observed $\alpha_{i'} \approx \frac{e^{z_{i*}}}{\beta_{i*}e^{z_{i*}}} = \frac{1}{\beta_{j*}} > 1$. Consequently, the observed probability $p_{i'}$ is dominated by $\beta_{j*}$ so that $\alpha_{i'} > 1$. By applying Claim 1, when the peak probability does not correspond to the positive sample class, this induces a decrease in the peak probability while simultaneously increasing all other class probabilities.

### A.2 PROOF OF THEOREM 3.1

**Proof.**

To prove Theorem 3.1, we need to prove $\frac{E_P[e^T]}{E_Q[e^T]} \leq E_Q[(\frac{dP}{dQ})^2]$:

We first denote $f = \frac{e^T}{E_Q[e^T]}$, then obviously we can get

$$\begin{cases} E_Q[f] = 1, \\ \frac{E_P[e^T]}{E_Q[e^T]} = E_P[f] \end{cases} \tag{12}$$

Our objective is to demonstrate that the maximum value of $\frac{E_P[e^T]}{E_Q[e^T]} = E_P[f]$ is $E_Q[(\frac{dP}{dQ})^2]$, next we only need to find the the maximum value of $E_P[f]$. Under the constraint of $E_Q[f] = 1$, find the maximum value of $E_P[f]$, using the Lagrange multiplier method, let

$$\mathcal{L}_f(v) = E_P[f] - \lambda(E_Q[f] - 1) \tag{13}$$

$$= \int p(v)f(v)dv - \lambda\left(\int q(v)f(v)dv - 1\right) \tag{14}$$

Setting the derivative of $\mathcal{L}$ to zero, we can get

$$\frac{\delta \mathcal{L}}{\delta f(v)} = p(v) - \lambda q(v) = 0 \implies f(v) = \frac{p(v)}{\lambda q(v)} \tag{15}$$

Substituting into the constraint $\int q(v) f(v) dv = 1$, we can get

$$\int q(v) \cdot \frac{p(v)}{\lambda q(v)} dv = 1 \tag{16}$$

$$\implies \frac{1}{\lambda} \int p(v) dv = 1 \tag{17}$$

$$\implies \frac{1}{\lambda} = 1 \tag{18}$$

$$\implies \lambda = 1 \tag{19}$$

Substituting into the constraint $f(v) = \frac{p(v)}{\lambda q(v)}$, the maximum of $E_P[f]$ is :

$$\frac{E_P[e^T]}{E_Q[e^T]} = E_P[f] \leq \int p(v) \frac{p(v)}{q(v)} dv \tag{20}$$

$$= \int p(v) \frac{p(v)}{q(v)} \frac{q(v)}{q(v)} dv \tag{21}$$

$$= E_Q[(\frac{dP}{dQ})^2] \tag{22}$$

Taking the logarithm of both sides, we conclude the proof of the Theorem 3.1.

### A.3 PROOF OF STABILITY OF VARIATIONAL GAP

**Proof.**

We first denote that $\mathcal{J} = -\mathcal{L}_{PFO}$.

We need to prove that the resulting variational gap $\mathcal{D}_2 - \mathcal{J}$ gradually decreases during training until the optimization objective approaches the upper limit of the parameterization capability.

To prove that the gap is stable, we construct a Lyapunov function:

$$V_\theta = \mathcal{D}_2(\mathcal{P}||\mathcal{Q}) - \mathcal{J} \geq 0 \tag{23}$$

The derivative is given by:

$$\frac{dV}{dt} = -\nabla_\theta \mathcal{J} \cdot \frac{d\theta}{dt} \tag{24}$$

$$= -\nabla_\theta \mathcal{J} \cdot (\eta \nabla_\theta J) \tag{25}$$

$$= -\eta ||\nabla_\theta \mathcal{J}||^2 \leq 0 \tag{26}$$

Lyapunov stability theory guarantees that $V_\theta$ converges to a minimum, ensuring that the training process minimizes $V_\theta$ until $||\nabla_\theta J|| = 0$.

Furthermore, we show that the upper bound of the gap is on the order of $O(\sqrt{\mathcal{D}_2})$.

$$V_\theta = log\frac{E_Q[(\frac{dp}{dq})^2]E_Q[e^T]}{E_P[e^T]} \tag{27}$$

$$= log\frac{E_Q[(\frac{dp}{dq})^2]E_Q[e^T]}{E_Q[\frac{dp}{dq}e^T]} \tag{28}$$

$$= log\frac{E_Q[r^2]E_Q[e^T]}{E_Q[re^T]} \tag{29}$$

Here, $r$ denotes the likelihood ratio. We next employ the covariance to decompose the denominator.

$$E_Q[re^T] = E_Q[r]E_Q[e^T] + Cov(r, e^T) \tag{30}$$

Substituting $E_Q[r] = 1$, we have

$$V_\theta = log\frac{E_Q[r^2]E_Q[e^T]}{E_Q[e^T] + Cov(r, e^T)} \tag{31}$$

$$= log\frac{E_Q[r^2]}{1 + \frac{Cov(r,e^T)}{E_Q[e^T]}} \tag{32}$$

From the Cauchy–Schwarz inequality, we have

$$|Cov(r, e^T)| \leq \sqrt{Var(r) \cdot Var(e^T)} \tag{33}$$

During actual optimization, $e^T$ is amplified for positive pairs and suppressed for negative ones. This process drives a positive correlation between $e^T$ and the reward $r$, specifically resulting in a positive covariance.

The condition $Cov(r, e^T) > 0$ allows us to infer that

$$V_\theta \leq log\frac{E_Q[r^2]}{1 + \frac{\sqrt{Var(r) \cdot Var(e^T)}}{E_Q[e^T]}} \tag{34}$$

It is crucial to note that $T$ is constrained to the interval $(0, 1)$ in our parameterization. Given this constraint, it follows from Popoviciu's inequality that

$$Var(e^T) < \frac{1}{4}(e^1 - e^0)^2 = (\frac{e-1}{2})^2 \tag{35}$$

It is noteworthy that $Var(r) = E_Q[r^2] - 1$, which stems from the fact that $E_Q[r] = 1$. Thus we can get

$$V_\theta < log\frac{E_Q[r^2]}{1 + \frac{(\frac{e-1}{2})\cdot\sqrt{E_Q[r^2]-1}}{E_Q[e^T]}} \tag{36}$$

$$< log\frac{E_Q[r^2]}{1 + \frac{e-1}{2e} \cdot \sqrt{E_Q[r^2] - 1}} \tag{37}$$

The gap is bounded by a term of order $log\sqrt{E_Q[r^2]}$, i.e., $log\sqrt{\mathcal{D}_2}$, which completes the proof.

# B  EXPERIMENTAL DETAILS

## B.1  IMPLEMENTATION DETAILS

**Training Hyperparameters** For general hyperparameters, we applied the following setup: We used a batch size of 128, a maximum sequence length of 8192, and a cosine learning rate schedule with a 10% warmup step per epoch, with all implemented using the Adam optimizer (Adam et al., 2014). The meaning of the hyperparameter $\beta$ varied across the different baseline methods, and the corresponding search strategies are shown in Table 3. Each method independently searched for a learning rate in the range [5e-7, 8e-7, 1e-6]. For the binary preference learning algorithm, the positive and negative weights can be separated and adjusted separately. In this paper, the default value $\lambda_l = \lambda_w = 1.0$.

Table 3: Hyperparameter Search Range for Different Methods

| Method | Hyperparameter Search Range |
|--------|------------------------------|
| KTO | $\begin{cases} \lambda_l = \lambda_w = 1.0 \\ \beta \in [0.01, 0.05, 0.1] \end{cases}$ |
| KLDO | $\begin{cases} \lambda_l = \lambda_w = 1.0 \\ \beta \in [0.01, 0.05, 0.1] \end{cases}$ |
| PFO+ | $\begin{cases} \lambda_l = \lambda_w = 1.0 \\ \beta \in [0.01, 0.05, 0.1] \end{cases}$ |
| PFO | $\begin{cases} \lambda_l = \lambda_w = 1.0 \\ \beta \in [0.01, 0.05, 0.1] \end{cases}$ |

## B.2  RECOMMENDATION PROMPT

| |
|---|
| **Interaction history:** Independence Day (a.k.a. ID4) (1996),Star Wars: Episode VI - Return of the Jedi (1983),True Lies (1994),Aladdin (1992),Fargo (1996) |
| **Recommendation Prompt Example:**
**Instruction:** Using the past movie choices, can you predict the next they will likely watch?
Independence Day (a.k.a. ID4) (1996),Star Wars: Episode VI - Return of the Jedi (1983),True Lies (1994),Aladdin (1992)
**output:** Fargo (1996) |

Table 4: Example of Interaction History and Recommendation Prompt

Note that during actual evaluation, we will ignore matching the content inside parentheses, but we will not perform any item name validity checks to filter out invalid recommendations.

# C  LLM USAGE

We used a large language model to polish the translation of the article, but we carefully checked that the modifications did not change the content of the original text, but only changed the quality of the translation.

