# OpenReview forum: "PFO: Optimizing binary Preference Alignment from a Probability Flow Perspective"
_ICLR.cc/2026/Conference — Submitted to ICLR 2026_

### Official Review · Reviewer_okdM · 2025-10-29

**Soundness:** 2
**Presentation:** 1
**Contribution:** 1
**Rating:** 2
**Confidence:** 3

**Summary:**

This paper suggests a potential squeezing effect in binary preference alignment (like KTO), where optimizing on single positive or negative samples can lead to suboptimal probability distributions. Specifically, it suggests that probability mass from penalized negative samples may flow to neutral non-target regions, and mass for rewarded positive samples may be drawn from these same neutral regions, rather than directly from the negative ones.

To address this, the authors propose Probability Flow Optimization (PFO), a new loss function that reweights samples to encourage a more direct flow of probability from negative to positive distributions. The core modification is the application of a sigmoid function to the log-probability ratio ($r_{\theta}$) before it is used in the loss. The authors claim this stabilizes training by preventing "sample drowning" and gradient explosion from unbounded $r_{\theta}$ terms (as seen in KLDO), thereby allowing the reweighting to work effectively. The paper presents PFO as a variational lower bound on the Rényi $\mathcal{D}_2$ divergence.

**Strengths:**

- PFO and PFO+ demonstrate consistently strong performance, outperforming previous baselines (KTO, KLDO) across multiple models (Llama3, Gemma2) and on two different task domains (dialogue and recommendation), although the margins are small.

- The final algorithm is simple to implement, essentially modifying the KLDO loss by wrapping the $r_{\theta}$ term in a sigmoid function.

- The inclusion of the PFO+ variant is useful. It helps isolate the contribution of the sigmoid function (as PFO+ is nearly identical to KLDO but with $\sigma(r_{\theta})$)

**Weaknesses:**

- I think the central weakness of this paper is the significant gap between its main claim (solving the squeezing effect) and its evidence. The squeezing effect is defined as probability mass flowing incorrectly to/from "neutral regions" or "non-target samples". However, the paper never measures this probability flow. While a larger margin is good, it is not direct proof that the squeezing effect (as defined) has been solved. The results demonstrate that PFO is a more effective optimizer for separating $y_w$ and $y_l$, but they fail to show why. The improved performance could be due to other factors, such as general gradient stabilization, rather than a specific correction of probability flow.

- The core argument for why the sigmoid function is the correct solution is convoluted. The logic appears to be: (1) The squeezing effect would be mitigated by reweighting samples. (2) Existing reweighting (like KLDO's $e^{r_{\theta}}$) is unstable and causes "sample drowning." (3) The sigmoid $\sigma(r_{\theta})$ stabilizes this reweighting. This is an indirect justification. Additionally, I found the paper writing to be often unclear, with some typos and inconsistencies ("off-line" rather than "offline" on lines 39 and 112, "Pairwise" capital on line 43) as well as unclear sentences (line 86 "where positive data require more protection" -> why?, "data acquisition because online feedback is often directed at a displayed reply and either likes or dislikes it" -> grammatically incorrect and I could mention more)

- The primary technical contribution is the addition of a sigmoid function to the KLDO loss. This could be interpreted as a minor stabilisation trick (i.e., a form of gradient clipping) rather than a novel alignment algorithm. Given that PFO+ (which performs very well) is only this modification, I don't see the novelty of the contribution.

**Questions:**

- Can the authors provide more direct evidence to support their central claim? For example, could they design an experiment to measure the probability mass assigned to a set of "neutral" or "non-target" samples and show that PFO, unlike KTO or KLDO, successfully prevents probability from flowing to this set?

- The justification for the sigmoid is to prevent "sample drowning" from the $e^{r_{\theta}}$ term in KLDO. Can the authors provide an analysis (e.g., gradient statistics, or the contribution of the top-k samples to the batch gradient) to prove that this is a significant problem in practice for KLDO, and that the sigmoid function effectively mitigates it?

---

> ### Author Response · Authors · 2025-11-18
>
> Thank you for reviewing our paper. However, you may have some misunderstandings about our work, so we would like to clarify them here.
>
> First, we want to clarify that our work is not simply about adding a sigmoid function. After further analysis of the compression effect, we conclude that compression should be avoided as much as possible during training on both positive and negative samples. Based on this, we propose the PFO algorithm. This part can be found in sections 3.2 and 3.3 of the paper, where we provide a more thorough theoretical analysis of the compression principle and propose the corresponding PFO algorithm. Gradient analysis, divergence analysis, and stability analysis are presented in sections 3.4.1 and 3.4.2.
>
> Furthermore, we find that this perspective allows for a new interpretation of the KLDO algorithm, as it optimizes only for the compression effect on negative samples. This better protects positive samples compared to schemes that optimize for compression effects on both positive and negative samples. Therefore, in these scenarios where "prioritizing sufficient reward and protection for positive samples" is prioritized, we degenerate the PFO algorithm, simplifying it to PFO positive, to address this type of application.
>
> We point out that our PFO positive algorithm still differs from KLDO in this scenario. This is because KLDO's weighting method can lead to sample overload. Even when our loss function degenerates into PFO positive, which optimizes only the negative direction, it still differs from KLDO by adding a sigmoid function. This sigmoid function significantly improves the stability of weighted sampling and training efficiency.
>
> Our experimental results show that the bilaterally optimized PFO outperforms KTO and KLDO in general scenarios, and PFO positive performs exceptionally well in scenarios where "positive samples are far more important than negative samples."
>
> **In summary, our work not only adds another sigmoid to the KLDO loss, but also provides a more detailed theoretical analysis of the squeezing principle in binary preference alignment scenarios and proposes a weighted approach to alleviate squeezing. If both positive and negative samples are weighted, it becomes the PFO algorithm; if weighting is done only in the negative direction like KLDO, it becomes the PFO+(positive) algorithm. We demonstrate through experiments that this weighting method yields benefits in both general and recommendation domains.**
>
> Regarding your question:
>
> Regarding the probability monitoring of non-target regions you mentioned: Our goal is to maximize the probability flow by avoiding interference from non-target regions, thereby maximizing the inflow of positive samples and the outflow of negative samples. Therefore, this core objective can be mathematically transformed into log_chosen - log_rejected. Thus, we do not perform additional probability monitoring on non-target regions.
>
> Regarding the necessity of the sigmoid function you mentioned: As I mentioned before, our work does not merely focus on the differences in the sigmoid function itself. We only theoretically explain why the sigmoid function is introduced into the PFO and PFO+ loss designs. We also prove that in the PFO+ version considering only negative compression optimization, introducing the sigmoid function is indeed superior to KLDO.
>
> Looking forward to your reply and suggestions.

---

> ### Author Response · Authors · 2025-11-28
>
> Dear Reviewer okdM,
>
> We sincerely appreciate the time and effort you've devoted to reviewing our work. We understand that your schedule may be quite busy. As the authors-reviewer discussion phase draws to a close, we kindly request your attention to our responses. Our aim is to gain insights into whether our responses effectively address your concerns and to ascertain if there are any additional questions or points you would like to discuss.

---

### Official Review · Reviewer_yLfu · 2025-11-01

**Soundness:** 2
**Presentation:** 3
**Contribution:** 3
**Rating:** 6
**Confidence:** 3

**Summary:**

The paper proposes Probability Flow Optimization (PFO) as an alternative objective for training on pairwise examples. The paper begins by extending the squeezing effect claims of Ren & Sutherland 2024 to additionally account for the effect of positive training (as prior work only looked at negative training). The paper then proposes PFO, motivated by the goal of reducing the squeezing effect. The paper compares the gradient of the proposed method to those of KTO and KLDO conceptually, and proves that PFO is a lower bound for the Renyi divergence ($\alpha=2$) between positive and negative samples. Experiments show that PFO increases log probability margins more than baseline methods and yields better results in a dialogue and recommendations experiment respectively.

**Strengths:**

1. The paper builds on impactful analysis around the squeezing effect introduced in Ren & Sutherland 2024, offering a more holistic picture the probability changes of tokens between gradient steps by adding the analysis for positive training.
2. The relationship between the objective and the Renyi divergence, $\alpha=2$, is interesting and offers useful intuition, and the stability proof builds confidence that it is not vacuous.
3. The insight provided by the lines 216-229 and Figure 3 are a nice detail in describing the development of the method.
4. The authors test PFO in multiple experimental settings and show positive downstream performance.

**Weaknesses:**

1. While the motivation for the method is overcoming the squeezing effect and enabling more probability flow from negative to positive examples, the experiments still show that PFO suffers from log probabilities going down for both chosen and rejected. Indeed, even though PFO yields the largest log prob differences between chosen and rejected, Figure 2 shows that chosen and rejected log probs still go down (and in fact the most) with the proposed method. Thus, the connection between the squeezing effect analysis and the proposed PFO method seems weak.
2. PFO makes quite a few different changes relative to existing methods, and it is hard to ascertain / isolate the influence of each. Either additional experimental ablations or a more step-by-step walkthrough of the design choices would help build confidence in the method and the insights ascertained.
3. The gradient analysis of 3.4.1 seems a bit superficial, including claims such as the lack of an adaptive weighting may cause situations in Claim 3 and Extended Claim 3. This section would be be much more impactful with experimental evidence.
4. The experiments would benefit from analyses showcasing what differences are significant.

**Questions:**

1. Could the authors clarify the design choices for the PFO objective in a more step-by-step fashion?
2. What is the significance of the log prob difference results (i.e, PFO achieving the largest) when chosen and rejected log probas still go down (and the most) for the proposed method?
3. Could the authors connect the hypotheses made in the gradient analysis with experiments?
4. Could the authors conduct significance testing for the experimental results?

---

> ### Author Response · Authors · 2025-11-24
> **Response 1**
>
> Thank you very much for your careful review and feedback!
>
> First, following your suggestions, we added some experiments. We demonstrated that the risk of decreased positive sample probability and increased negative sample probability does exist, as we mentioned, and showed that our loss function can mitigate this risk. The specific experimental setup and results are as follows:
>
> We trained the model using KTO and PFO for either negative-only (PFO-positive) or positive-only (PFO-negative) training. The naming is to align with the paper's PFO+ (PFO-positive), which incorporates optimization in the negative training to mitigate the decrease in positive sample probability. We monitored the positive probability (not trained) during PFO-positive training and the negative probability (not trained) during PFO-negative training.
>
> According to our theoretical expectation, although PFO-positive only underwent negative training, its positive sample probability is at risk of decreasing, while PFO-negative's negative sample probability is at risk of increasing.
>
> The experimental results are as follows:
>
> KTO-negative-logp-rejected
> [-1.1934, -1.1792, -1.1758, -1.1675, -1.1523, -1.1626, -1.1543, -1.1641, -1.1543, -1.1431, -1.1436, -1.1323, -1.1372, -1.1401, -1.1343, -1.1372, -1.1274, -1.1348, -1.126, -1.1274, -1.1245, -1.125, -1.1191, -1.1167, -1.1191, -1.1157, -1.1125, -1.1121, -1.1108, -1.1135, -1.1101, -1.113, -1.1123, -1.1108, -1.1067, -1.1067, -1.106]
> PFO-negative-logp-rejected
> [-1.1938, -1.1821, -1.1738, -1.165, -1.1523, -1.1626, -1.1592, -1.1631, -1.1528, -1.1426, -1.1465, -1.1357, -1.1404, -1.1392, -1.134, -1.1406, -1.1284, -1.1362, -1.1255, -1.127, -1.1289, -1.127, -1.1211, -1.1199]
> KTO-positive-logp-chosen
> [-52.1875, -58.5312, -59.5, -59.5, -59.7812, -59.8125, -60.0312, -60.125, -60.1562, -60.1875, -60.1562, -60.1562, -60.1562, -60.1875, -60.1406, -60.2344, -60.1094, -60.2344, -60.2031, -60.1562, -60.1719, -60.2344, -60.2031, -60.2344, -60.2344, -60.2344, -60.2344, -60.2344]
> PFO-positive-logp-chosen
> [-49.8906, -58.9062, -59.5, -59.5312, -59.8125, -60.0, -60.0781, -60.1562, -60.1094, -60.1406, -60.2031, -60.1719, -60.1875, -60.2031, -60.1094, -60.2031, -60.1875, -60.2188, -60.2188, -60.1875, -60.2344, -60.2188, -60.2344, -60.2344, -60.2344, -60.2031, -60.2344]
>
> Negative-logp-rejected
> Results of significant differences
> pvalue=0.0301534332917288
> KTO_mean vs PFO_mean -1.1330170502533783 vs -1.1457417805989583
>
> Positive-logp-chosen
> Results of significant differences
> pvalue=0.8795489670675534
> KTO_chosen vs PFO_chosen -59.76171875 vs -59.689236111111114
>
> 1. It can be seen that when only positive training is performed, rejected samples continuously increase; when only negative training is performed, chosen samples continuously decrease, which is consistent with our theoretical analysis of the squeezing principle.
> 2. For both the training data and the trained model, the probability of negative samples is much greater than the probability of positive samples; the training data is not an evenly sampled result.
> 3. Both positive and negative squeezing optimization yield benefits in mitigating squeezing. The benefit to chosen samples is not significant, possibly because the negative training samples themselves are too numerous, which is consistent with phenomenon 1.

---

> > ### Author Response · Authors · 2025-11-24
> > **Response 2**
> >
> > Q1: Our design is based on the theory of squeezing; we hope to add a reweighting mechanism to both positive and negative samples. Furthermore, we observed the KTO loss and found in actual training that the KL margin in the KTO sigmoid is often ineffective (because it takes the maximum value with 0, and pi_theta is usually less than pi_ref), so we removed the KL margin operation here. Combining these two points, we designed the form logE[e^r]. We also realized that a sigmoid function is needed to protect samples, preventing sample overload and gradient stability issues. Finally, we found that using only pi_theta as r affects the weighting effect due to the range of values, so we added pi_ref to broaden the range. Finally, we realized that KLDO can also be interpreted from the perspective of the squeezing principle, although this was not the author's intention. However, because the sigmoid design enhances training efficiency, this aligns with our original intention in designing the sigmoid function.
> >
> > Q2: We understand that during training, some weighting settings weaken the contribution of some samples. While this alleviates the corresponding squeezing, it may also lead to the sacrifice of the direct contribution of some samples. As described in Phenomenon 2 of our supplementary experiment, the probabilities of positive and negative samples are not equal. This coupling leads to a significant decrease in the log probability of PFO, despite the largest difference in log probabilities. Therefore, in scenarios where positive samples are more significant than negative samples, we supplemented the FPO+ scheme, which offers a better trade-off in these situations (as seen in actual experimental results in the recommendation field). Specifically, we will explore the trade-off between this mitigation of the squeeze effect and the direct training effect in future work.
> >
> > Q3 & Q4 I apologize that I didn't quite understand the connection between gradient analysis and experimental results you mentioned. Do our newly added experimental phenomena (with significant difference tests) prove our gradient analysis in the squeeze principle?
> >
> > Looking forward to your suggestions and replies.

---

> ### Author Response · Authors · 2025-11-28
>
> Dear Reviewer yLfu,
>
> We sincerely appreciate the time and effort you've devoted to reviewing our work. We understand that your schedule may be quite busy. As the authors-reviewer discussion phase draws to a close, we kindly request your attention to our responses. Our aim is to gain insights into whether our responses effectively address your concerns and to ascertain if there are any additional questions or points you would like to discuss.

---

### Official Review · Reviewer_utQP · 2025-11-01

**Soundness:** 3
**Presentation:** 3
**Contribution:** 3
**Rating:** 4
**Confidence:** 3

**Summary:**

The paper identifies that the squeezing effect in binary preference alignment affects both positive and negative samples training . They propose PFO which dynamically reweights samples to encourage “probability mass flow” from negative to positive distributions. The experimental section shows performance on Alpacaeval 2 and MT Bench. Overall this is a nice extension (to some degree) to the KLDO paper but has some structural inconsistencies as below.

**Strengths:**

1. The paper tackles an interesting problem and provides a simple solution, backed with some theoretical underpinning

2. The paper is well written in most places. Comprehensive and easy to follow.

**Weaknesses:**

1. The utility of PFO+ is a bit confusing to the main premise of the paper. On one hand the authors are motivating the paper at the start by saying that both negative and positive training suffer from the squeezing effect. They even have the extended claim to explicitly account for the problem with the positive training. Then they propose PFO+ which only handles a fix to the negative samples by adding a simple sigmoid over KLDO. Ideally it should still suffer from the problem. In Table 2 they show that PFO+ outperforms KLDO, but that's just on a very narrow recommendation domain, does it generalize, it feels very forcibly included? It's as if the authors propose that the problem lies with both positive and negative training and then go on to propose a method that just ignores the premise. Further PFO+ outperforms PFO in recsys seems like fixing positive sample squeezing effect is not at all always needed. PFO+ even once in a while does better even in the MT Bench or Alpacaeval which muddies the premise further.

2. Further in while PFO has the largest $\delta$ margin, it is unclear why or how KTO even with better margin than KLDO and PFO+ (except llama3 base, which is strange on its own) has worse results than both in the tables. It is strange.

3. The authors use ultrafeedback dataset annotated using the armorm reward model for preference optimization, the dataset already has chosen and rejected pairs of responses and even ranked responses. If achieving great results on MT Bench and Alpaca 2 is the goal, then they should just run some obvious pairwise feedback optimization algorithm on the same dataset to show how far off they are, something like at least DPO, and then perhaps SimPO or ORPO etc. as baselines. Very specifically since the paper builds a premise around binary feedback being cheaper. This omission makes it tricky to judge the real world value of the algorithm.
    The authors can argue why we would want to compare binary algorithms (KTO, PFO etc) with those like DPO, SimPO, ORPO etc. By using a pairwise dataset but only testing binary feedback algorithms the paper avoids answering the question: “what is the performance tradeoff for using the cheaper binary alignment?” moreover, both the baselines KTO and KLDO papers had comparisons with other pairwise preference methods that were SOTA at that time.

4. Unclear what happens with the reasoning models with PFO, for example if your base is a reasoning model? The reasoning tokens could be very similar leading to either right or wrong answer part of the response, i would imagine the log probs may still be very close between the positive and negative examples. In this case wouldn't the scores be similar too? could this lead to inconsistent reasoning behavior of the model itself?

**Questions:**

there are some open questions in the weaknesses section, if addressed, I consider increasing my score.

---

> ### Author Response · Authors · 2025-11-18
>
> Thank you for reviewing our paper. However, there may have been some misunderstandings regarding our work, so let us clarify our work here.
>
> 1. The Significance of PFO+. First, we state that our work mainly focuses on optimizing the mitigation of the squeezing effect. Based on this, we propose a weighted approach to alleviate squeezing, namely PFO, which optimizes simultaneously during positive and negative training. We then unexpectedly discovered that the KLDO approach can also be interpreted from this weighted squeezing mitigation perspective, although it wasn't initially designed with this idea in mind. They only optimize the negative squeezing effect, which may be meaningful for scenarios that protect positive samples. For example, in recommendation scenarios, the negative labels may not be completely negative. Therefore, we added some experiments. According to our loss design, with only half the optimization, it still yields higher returns compared to KLDO. We also theoretically explained the core difference, which lies in the necessity of the sigmoid function during weighting. As you mentioned, recommendation is indeed a very narrow field, a scenario where "the label meaning of positive samples is far greater than that of negative samples." It is in this scenario that PFO+ performs better, while the effectiveness of PFO is verified on the more generalized general domains MT-Bench and Alpaca2. To summarize, we present two different solutions to mitigate the squeezing effect for two different scenarios.
>
> 2. Why does KTO, with its larger margin, perform worse? The larger margin you mentioned in KTO refers to the experimental figures in Figures 2 and 5, both for the more generalized general domain. It can be seen that on MT-Bench and Alpaca2, KTO scores better than PFO+ and KLDO. From the perspective of the squeeze mitigation we proposed, KLDO and PFO+ exist specifically for positive samples where the label significance is greater than that of negative samples, while negative samples in the general domain are equally important.
>
> 3. Comparison with algorithms like DPO. You mentioned "since the paper builds a premise around binary feedback being cheaper. This omission makes it tricky to judge the real-world value of the algorithm." Here, I want to explain that the lower cost we refer to is the cost of data acquisition. Because in many online scenarios, it is not easy to obtain paired feedback, and liking or disliking a single reply is more in line with user intuition. In this case, due to data deficiencies, paired training like DPO may not be feasible, hence the exploration of binary preference training. This is precisely why we didn't include DPO in the comparison, as it's impossible to use DPO for comparison in real-world scenarios. We used this dataset simply because it's authoritative and widely known. During training, we also separated the positive and negative labels, creating datasets with only positive or only negative labels.
>
> 4. If I understand correctly, you're asking about the possibility that if the model is an inference model, the token probabilities in the inference process might be similar for both positive and negative samples. This is an excellent question. Inference models are indeed a scenario we haven't considered at this stage. We will explore the performance and optimization strategies for this scenario in the future.
>
> Looking forward to your reply.

---

> ### Author Response · Authors · 2025-11-28
>
> Dear Reviewer utQP,
>
> We sincerely appreciate the time and effort you've devoted to reviewing our work. We understand that your schedule may be quite busy. As the authors-reviewer discussion phase draws to a close, we kindly request your attention to our responses. Our aim is to gain insights into whether our responses effectively address your concerns and to ascertain if there are any additional questions or points you would like to discuss.

---

> ### Comment · Reviewer_utQP · 2025-11-28
>
> I thank the authors for the clarifications. But I do not feel they substantially resolve the key conceptual and methodological concerns I raised. Given this, I will keep my current score.

---

> > ### Author Response · Authors · 2025-11-29
> >
> > Dear Reviewer utQP,
> >
> > In more detail, what are the key concepts and methodologies you mentioned that were not resolved?
> >
> > Let us briefly summarize your question and our response:
> >
> > 1. You mentioned concerns about the practicality of PFO+ and PFO.
> >
> > In our paper and response, we stated that PFO+ focuses on scenarios where "positive samples need more protection," while PFO covers general-domain scenarios. Both solutions are derived from our proposed theory of mitigating crowding, and their effectiveness has been verified in both scenarios.
> >
> > 2. You mentioned that KTO's margin is larger than KLDO and PFO+, but its performance is worse.
> >
> > This is a misunderstanding. KTO's larger margin compared to KLDO and PFO+ is a result in the general domain, and it has been proven that KTO performs better than KLDO and PFO+ in the general domain.
> >
> > 3. You asked why we didn't compare the DPO algorithm.
> >
> > Because DPO is a pairwise optimization algorithm, while we mainly target binary preference scenarios, where paired samples do not exist. We use the UltraFeedback dataset because it is a sufficiently authoritative dataset, and we train it by splitting it into binary form.
> >
> > 4. You mentioned how PFO would perform if the model were an inference model.
> >
> > We primarily focus on offline preference alignment in binary scenarios, and reasoning models are not well-suited for this training method; therefore, inference models are not currently within our consideration.

---

### Author Response · Authors · 2025-11-27
**Added some experiment**

We added some experiments. We demonstrated that the risk of decreased positive sample probability and increased negative sample probability does exist, as we mentioned, and showed that our loss function can mitigate this risk. The specific experimental setup and results are as follows:

We trained the model using KTO and PFO for either negative-only (PFO-positive) or positive-only (PFO-negative) training. The naming is to align with the paper's PFO+ (PFO-positive), which incorporates optimization in the negative training to mitigate the decrease in positive sample probability. We monitored the positive probability (not trained) during PFO-positive training and the negative probability (not trained) during PFO-negative training.

According to our theoretical expectation, although PFO-positive only underwent negative training, its positive sample probability is at risk of decreasing, while PFO-negative's negative sample probability is at risk of increasing.

The experimental results are as follows:

KTO-negative-logp-rejected [-1.1934, -1.1792, -1.1758, -1.1675, -1.1523, -1.1626, -1.1543, -1.1641, -1.1543, -1.1431, -1.1436, -1.1323, -1.1372, -1.1401, -1.1343, -1.1372, -1.1274, -1.1348, -1.126, -1.1274, -1.1245, -1.125, -1.1191, -1.1167, -1.1191, -1.1157, -1.1125, -1.1121, -1.1108, -1.1135, -1.1101, -1.113, -1.1123, -1.1108, -1.1067, -1.1067, -1.106]
PFO-negative-logp-rejected [-1.1938, -1.1821, -1.1738, -1.165, -1.1523, -1.1626, -1.1592, -1.1631, -1.1528, -1.1426, -1.1465, -1.1357, -1.1404, -1.1392, -1.134, -1.1406, -1.1284, -1.1362, -1.1255, -1.127, -1.1289, -1.127, -1.1211, -1.1199]
KTO-positive-logp-chosen [-52.1875, -58.5312, -59.5, -59.5, -59.7812, -59.8125, -60.0312, -60.125, -60.1562, -60.1875, -60.1562, -60.1562, -60.1562, -60.1875, -60.1406, -60.2344, -60.1094, -60.2344, -60.2031, -60.1562, -60.1719, -60.2344, -60.2031, -60.2344, -60.2344, -60.2344, -60.2344, -60.2344]
 PFO-positive-logp-chosen [-49.8906, -58.9062, -59.5, -59.5312, -59.8125, -60.0, -60.0781, -60.1562, -60.1094, -60.1406, -60.2031, -60.1719, -60.1875, -60.2031, -60.1094, -60.2031, -60.1875, -60.2188, -60.2188, -60.1875, -60.2344, -60.2188, -60.2344, -60.2344, -60.2344, -60.2031, -60.2344]

Negative-logp-rejected Results of significant differences pvalue=0.0301534332917288
KTO_mean vs PFO_mean -1.1330170502533783 vs -1.1457417805989583
Positive-logp-chosen Results of significant differences pvalue=0.8795489670675534
KTO_chosen vs PFO_chosen -59.76171875 vs -59.689236111111114

1. It can be seen that when only positive training is performed, rejected samples continuously increase; when only negative training is performed, chosen samples continuously decrease, which is consistent with our theoretical analysis of the squeezing principle.
2. For both the training data and the trained model, the probability of negative samples is much greater than the probability of positive samples; the training data is not an evenly sampled result.
3. Both positive and negative squeezing optimization yield benefits in mitigating squeezing. The benefit to chosen samples is not significant, possibly because the negative training samples themselves are too numerous, which is consistent with phenomenon 1.

---

### Meta-Review · Area_Chair_Z3s5 · 2025-12-27

**Summary:**

While the reviewers agree that PFO has quite a few differences with the existing methods (e.g., KTO, KLDO), they unanimously believe that there is a significant gap between the main claim of the paper, namely solving the squeezing effect, and the evidence provided by the authors. There is no theory and neither enough empirical results to show that log probabilities for both chosen and rejected responses do not go down in PFO -- or in other words PFO addresses the squeezing effect and prevents probability mass flowing incorrectly.

Despite its similarity to the existing methods, PFO makes quite a few changes relative to them. However, there are not enough ablation studies to measure the effect of each of these changes. I believe the authors could do a better job in explaining/analyzing their experimental results. The reviewers found some of these results surprising, and thus, more discussion or perhaps further experiments are needed for clarification.

Overall, the reviewers found the problem studied in the paper interesting and the method relatively novel, but they are not convinced that the proposed method delivers all the claims and many results remained unexplained. Thus, I would recommend the authors to take the reviewers' comments into account, improve their work and prepare it for future venues.

**Reviewer Concerns:**

I do not think that the authors managed to address the main concerns of the reviewers, which are mostly related to the significant gap between the main claim of the paper, i.e., solving the squeezing effect, and the evidence provided by the authors.

**Reviewer Scores:**

It would be hard to see how Reviewers yLfu and okdM would have changed their scores if they had participated in the discussion, but Reviewer utQP did not find the authors' initial response convincing to raise their score.

---

### Decision · Program_Chairs · 2026-01-26

Reject